# Primary Mucinous Cystadenocarcinoma of the Breast Intermixed with Pleomorphic Invasive Lobular Carcinoma: The First Report of This Rare Association

**DOI:** 10.3390/jpm13060948

**Published:** 2023-06-03

**Authors:** Federica Vegni, Nicoletta D’Alessandris, Angela Santoro, Giuseppe Angelico, Giulia Scaglione, Angela Carlino, Damiano Arciuolo, Michele Valente, Stefania Sfregola, Maria Natale, Alejandro Martin Sanchez, Valeria Masciullo, Gian Franco Zannoni, Antonino Mulè

**Affiliations:** 1Pathology Unit, Department of Woman and Child’s Health and Public Health Sciences, Fondazione Policlinico Universitario Agostino Gemelli IRCCS, 00168 Rome, Italy; federicavegni@gmail.com (F.V.); nicoletta.dalessandris@policlinicogemelli.it (N.D.); giulia.scaglione@policlinicogemelli.it (G.S.); angela.carlino@guest.policlinicogemelli.it (A.C.); damiano.arciuolo@policlinicogemelli.it (D.A.); dr.valente.m@gmail.com (M.V.); stefania.sfregola@guest.policlinicogemelli.it (S.S.); gianfranco.zannoni@unicatt.it (G.F.Z.); antonino.mule@policlinicogemelli.it (A.M.); 2Department of Medical and Surgical Sciences and Advanced Technologies G. F. Ingrassia, Anatomic Pathology, University of Catania, 95123 Catania, Italy; giuangel86@hotmail.it; 3Multidisciplinary Breast Center, Dipartimento Scienze della Salute della Donna e del Bambino e di Sanità Pubblica, Fondazione Policlinico Universitario Agostino Gemelli IRCCS, Largo Agostino Gemelli 8, 00168 Rome, Italy; maria.natale@policlinicogemelli.it (M.N.); martin.sanchez@hotmail.it (A.M.S.); 4Division of Gynecologic Surgery, Department of Woman, Child and Public Health, Fondazione Policlinico Universitario Agostino Gemelli IRCCS, Catholic University of Sacred Heart, 00168 Rome, Italy; valeria.masciullo@policlinicogemelli.it; 5Pathology Institute, Catholic University of Sacred Heart, 00168 Rome, Italy

**Keywords:** breast cancer, mucinous cystoadenocarcinoma, lobular carcinoma, mucinous carcinoma, immunohistochemistry, molecular landscape

## Abstract

Primary mucinous cystadenocarcinoma (MCA) is a rare breast carcinoma subtype showing overlapping histopathological features with mucinous cystadenocarcinoma of the ovary and pancreas. Current literature data suggest a favorable prognosis of breast MCAs despite its immunoprofile usually revealing lack of expression of estrogen receptor, progesterone receptor and HER-2 and high Ki67. As far as we know, only 36 cases have been reported in the literature to date. Its ambiguous morpho-phenotypic profile makes histological diagnosis very challenging. It must be distinguished from typical mucin-producing breast carcinomas and, above all, metastases from the same histotype in other sites (ovary, pancreas, appendix). Herein, we report the case of a primary breast MCA occurring in a 41-year-old female with peculiar histological features.

## 1. Introduction

Mucinous cystadenocarcinoma (MCA) is an extremely rare variant of invasive breast carcinoma usually presenting as a large (up to 19 cm) multicystic mass, arising in postmenopausal women (average 58 years old) [1,2,3,4,5,6,7,8,9,10,11,12,13,14,15,16,17,18,19,20,21,22,23,24,25,26,27,28,29,30,31]. As a distinct and well-documented entity, it has been introduced in the recent 5th edition of WHO classification of tumors of the breast, 2019 [32].

Tumors with overlapping morphological features, namely mucinous cystoadenocarcinomas, are more frequently described in the ovary, pancreas, and/or appendix [33]. First described by Koenig C. and Tavassoli in 1998, MCA of the breast is typically characterized by microscopic cystic spaces lined with bland-appearing tall columnar cells with abundant intracytoplasmic and extracytoplasmic mucin; stratification, tufting and papillary formations are also a common feature [2]. The degree of nuclear cytological atypia can vary, with focal loss of mucinous secretion and transformation in eosinophilic/squamoid cells [1,2,3,4,5,6,7,8,9,10,11,12,13,14,15,16,17,18,19,20,21,22,23,24,25,26,27,28,29,30,31]. Immunohistochemistry often results in a triple negative immunophenotype; however, the prognosis is generally favorable and nodal metastases are unusual [1,2,3,4,5,6,7,8,9,10,11,12,13,14,15,16,17,18,19,20,21,22,23,24,25,26,27,28,29,30,31]. Patients were followed up from 3 to 108 months, generally without reporting distant metastasis. Considering the morphological overlap with similar tumors of the ovary and gastrointestinal tract, establishing a diagnosis of MCA of the breast can be very challenging, especially on small bioptical samples [1,2,3,4,5,6,7,8,9,10,11,12,13,14,15,16,17,18,19,20,21,22,23,24,25,26,27,28,29,30,31]. 

To the best of our knowledge, less than 40 cases have been reported to date. Due to its extraordinary rarity, the pathogenesis and prognosis of this disease remain poorly understood. Moreover, a standard targeted treatment regimen is still lacking.

The aim of the present study is to enrich the scientific literature with another case diagnosed with this new emerging and intriguing entity arising in the breast of a 41-year-old female.

Clinico-pathological and immunohistochemical features as well as differential diagnosis and histogenesis are discussed in depth, reporting a detailed review of the literature on this topic.

## 2. Case Report

A 41-year-old woman was referred to our hospital for the presence of a painful, palpable nodule in her right breast. The patient had no history of hormonal replacement treatment or family history of cancer. Her menarche was at 13 years old. The woman mentioned a recent cesarean delivery of a healthy born. 

Mammography revealed a 15 mm solid focal asymmetric nodular density in the right internal equatorial location (Figure 1A). Ultrasound examination demonstrated the presence of a firm mass devoid of vascular structures, with an adjacent cyst with an overall diameter of 3 cm (Figure 1B). Serum levels of tumor markers were normal.

A core-needle biopsy was performed. The histological evaluation showed an invasive carcinoma consisting of two cells populations. The first one consisted of medium-sized cells, with large hyperchromatic nuclei, markedly pleomorphic, arranged in chains and/or single cells. Conversely, the second one consisted of tall columnar cells with basal-located hyperchromatic nuclei, and apical accumulation of intracytoplasmic mucin; nuclear degree of atypia varied from mild to moderate. These tall cells typically lined tortuous glands, often cystically dilated and with papillary formation. Immunohistochemistry (IHC) revealed different results in the two neoplastic populations. In detail, the first neoplastic component showed loss of E-cadherin and maintained expression of both estrogen (ER, 90%) and progesterone (PgR, 75%) receptors. Proliferation index (evaluated by Ki67) was about 15%. These results were consistent with the diagnosis of lobular invasive carcinoma (CLI) of the breast, pleomorphic variant. Conversely, mucinous atypical cells of the secondary neoplastic component showed diffuse immunoreactivity for pan-cytokeratin, CK7 and CK19; focal immunoreactivity for ER (5%, with bland staining), CK17, MUC1, MUC5AC and p16; and negative immunostaining for CK20, PgR, Androgen receptor (AR), HER2, GATA3, GCDFP15, CDX2, TTF1, NapsinA, WT1, PAX8, PSA. Elevated proliferation index (Ki-67), around 30%, was also documented. These results were consistent with the diagnosis of invasive carcinoma with mucinous features. Based on the morphological evidence of mucinous differentiation and the immunoreactivity for MUC1, MUC5AC, a provisional diagnosis of mixed invasive breast carcinoma with lobular and mucinous features was rendered. Moreover, a recommendation for clinicians to exclude the possibility of metastatic origin from the gastro-intestinal tract, pancreas and biliary tract was also included in the pathology report.

However, the patient’s clinical history was negative for previous neoplasms, and the esophagogastroduodenoscopy and colonoscopy were negative for neoplastic lesions in the digestive system. An ultrasound of the abdominal cavity was performed to rule out any occult primary in the adnexa or gastrointestinal tract. Moreover, a total body positron emission tomography (PET)/computed tomography (CT) was performed on the patient, revealing only the 16 × 10.5 × 6.5 mm heterogeneously enhancing breast tissue density lesion involving the right internal equatorial breast quadrant, without any other abnormality suggesting a possible metastatic origin. 

Therefore, breast lumpectomy with ipsilateral axillary lymph node dissection was subsequently performed. The surgical specimen confirmed the presence of a large irregular mass with a maximum diameter of 3 cm. At gross section, the tumor was grayish white, mainly nodular, partially cystic, with a honeycomb-like appearance, well-circumscribed, with pushing margins, and glistening gelatinous myxoid cut surface. The cyst walls were rough on palpation, and they were full of a pale yellow and jelly-like substance.

Overlying skin has not been involved. Histopathological examination showed an invasive carcinoma resulting from the mixture of two different neoplastic components (Figure 2). In detail, most of the neoplastic proliferation (about 75% of the whole tumor) consisted of variably sized interconnected mucin-filled cystic spaces lined with tall columnar atypical cells showing stratification, abundant intracytoplasmic mucin and basally located nuclei. Branched papillary or tufting excrescences extending and protruding into the cystic lumen have been frequently observed.

Nests of neoplastic cell floated in the mucin lake spaces were accompanied by moderate inflammatory cell infiltration.

The degree of cytological atypia was variable from mild to moderate. Neither marked nuclear pleomorphism nor high mitotic activity or large areas of tumoral necrosis have been observed. There was also a perilesional tiny focus of high nuclear grade ductal carcinoma in situ component with cribriform architecture. Microscopically, no clear evidence of myoepithelial layer was observed, and subsequent immunohistochemical results for myoepithelial markers immunostaining (p63, CK14 and Calponin) also confirmed the absence of a myoepithelial layer at the periphery of the cystic and papillary structures. No peripheral vascular invasion has been identified. The neoplasm contained very few inflammatory cells (Tumor Infiltrating Lymphocytes—TILs—about 5%). Skin and breast base were not histologically invaded. Surgical resection margins have been described as free. 

Moreover, neoplastic cells showed overlapping immunophenotype with the bioptical sample, namely positive immunostaining for ER (5%, with a weak, faint nuclear staining), CK7, CK19, CK20 (very focal staining), MUC1, MUC5AC and p16; and negative immunostaining for PgR, AR, HER2, GATA3, Mammaglobin, GCDFP15, p63, Ca19-9, CDX2, MUC6, SOX10, WT1, PAX8 and TTF1 (Figure 3). The positive site of E-Cadherin was located in the membrane of neoplastic cells. P53 immuno-pattern has been defined as being high wild-type. PDL-1 (Programmed cell Death Ligand 1 - clone sp142 Ventana Roche) was negative in immune cells (IC < 1%). Ki-67, as proliferation index, was about 35%. 

Our differential diagnosis based on location and peculiar morphology has included metastasis versus primary breast carcinoma. A broad immunohistochemical panel was performed to narrow a wide range of differential diagnoses: metastatic tumors from the ovaries or gastro-intestinal tract (including pancreas and appendix), but also different types of primitive breast carcinoma (mucinous carcinoma, encapsulated papillary carcinoma, invasive papillary carcinoma, cystic hypersecretory carcinoma and mucocele-like lesions).

The low estrogen receptor positive breast cancer profile helped us to rule out a mucinous carcinoma and an encapsulated papillary carcinoma, which typically diffusely express ER and PR. Additionally, breast mucinous carcinoma does not form cystic spaces; rather, it is composed of only small or larger clusters of epithelial neoplastic cells free-floating in extracellular mucin lakes. Nuclear grade in breast mucinous carcinoma is low to intermediate, while high-grade nuclei have been reported in few breast MCAs. 

We also ruled out the diagnosis of an invasive papillary carcinoma, which is usually composed of microcysts and papillary formations, but without intracellular and extracellular mucus; moreover, it is, generally, non-triple negative in biological phenotype. 

The possibility of a cystic hypersecretory carcinoma of the breast has also been excluded, because this other rare entity shows multiple variable-sized cystic spaces, containing colloid-like eosinophilic material and no intra/extracellular mucin.

Mucocele-like lesions of the breast, in which myoepithelium component is maintained around the cystically distended mucin-filled ducts, without any evidence of nuclear atypia, are benign in nature. The complete absence of myoepithelium in our neoplasia also ruled out this diagnosis. 

The above-mentioned morphological features along with the immunohistochemical profile were more consistent with the diagnosis of mucinous cystoadenocarcinoma, primitive of the breast. 

However, an additional neoplastic component, accounting for about 25% of the whole tumor, showed the classic morphological and immunophenotypical features of pleomorphic invasive lobular carcinoma of the breast: dispersed and dyscohesive cells with high nuclear grade, signet ring appearance, nuclear pleomorphism and loss of expression of E-cadherin. This neoplastic component retained expression of both estrogen (ER, 90%) and progesterone (PgR, 75%) receptors. Proliferation index (evaluated by Ki67) was about 15%. Her2 neu score was 0, according to ASCO/CAP 2018. 

Therefore, based on morphology, IHC and radiology, a final diagnosis of “mixed type primitive breast invasive carcinoma” with a prevalent (about 75%) Mucinous Cystoadenocarcinoma component and a minoritary (about 25%) pleomorphic invasive lobular carcinoma was rendered. 

The Nottingham grade was 2 (tubule formation = 2, nuclear pleomorphism = 2, and mitotic count = 2) in the prevalent mucinous component and 2 (tubule formation = 3, nuclear pleomorphism = 3, and mitotic count = 1) in the minority lobular component.

The biological profile was heterogeneous and different in the two neoplastic populations: low ER expressing (5%), PgR and Her2 neu negative, with high Ki67 value (35%) in the mucinous component; ER (90%) and PgR (75%) positive, Her2 neu negative, with low Ki67 value (15%), in the lobular component. 

Only 1 out the 13 removed axillary lymph nodes showed isolated tumoral cells (ITCs). The pathologic stage was pT2N0(i+)M0. 

No mutations in the BRCA1 and BRCA2 genes were identified in our case. According to a multidisciplinary tumor board decision, an adjuvant chemotherapy (4 cycles of docetaxel and cyclophosphamide) together with adjuvant radiotherapy has been proposed for the patient. From recent instrumental and clinical examinations, the patient is alive, in good health conditions, without evidence of recurrence and/or metastasis. A detailed timeline of clinical history of our patient is summarized in Figure 4.

## 3. Discussion

Compared to the corresponding histotypes arising in the ovary or pancreas, MCA of the breast is an exceedingly rare breast invasive carcinoma recently recognized as a separate neoplastic entity in the 2019 WHO classification of tumors of the breast [32]. It belongs to the family of mucin-producing carcinomas of the breast, constituting about 1–4% of total breast cancer and includes the following: mucinous carcinoma, mucinous micropapillary carcinoma, mucinous ductal carcinoma in situ, lobular carcinoma with extracellular mucin, solid papillary carcinoma, mucoepidermoid carcinoma and MCA [32]. To the best of our knowledge, only 36 cases of primary breast MCAs are currently reported in the literature [1,2,3,4,5,6,7,8,9,10,11,12,13,14,15,16,17,18,19,20,21,22,23,24,25,26,27,28,29,30,31]. The clinicopathologic features of all these cases, including the current one, are summarized in Table 1. They affect almost exclusively perimenopausal and postmenopausal women, with a median age of 72.5 years (range 41–96 years). The tumor size of the reported cases ranged from 0.8 cm to 19 cm, with two cases composed of multiple nodules. Breast MCA consists of multiple cystic spaces lined with a single or pseudostratified layer of tall columnar cells with basal nuclei, showing mild atypia, and an excess of intra/extracellular mucin [1,2,3,4,5,6,7,8,9,10,11,12,13,14,15,16,17,18,19,20,21,22,23,24,25,26,27,28,29,30,31]. In 19 cases, similarly to our case, an association with ductal carcinoma in situ (DCIS) was reported [1,2,3,4,5,6,7,8,9,10,11,12,13,14,15,16,17,18,19,20,21,22,23,24,25,26,27,28,29,30,31]. The co-existence of DCIS and MCA indicates that cancer cells of MCA may transform through mucinous metaplasia of epithelial cells of DCIS, accompanied by estrogen and progesterone receptors expression loss and that the in situ carcinoma form of MCA can derive from the mucinous metaplasia of epithelial ductal cells [5]. The morphological spectrum of MCA ranges from pure form of MCA to non-pure forms, represented by MCA with associated DCIS or MCA with both DCIS and invasive ductal carcinoma (IDC). Three cases have been described with focal squamous cell carcinoma differentiation and only one case with high-grade sarcomatoid component. The present case is the only one in which an association with an invasive pleomorphic lobular carcinoma was found. In this regard, intracellular or, more rarely, extracellular mucin, can be detected in a small subset of lobular carcinomas. However, in these cases, all the neoplastic cells are incohesive and lose E-Cadherin expression. 

As far as immunohistochemical molecular profile is concerned, most MCAs lack expression of ER, PR and Her2 neu, defining a triple negative phenotype. Focal and weak expression of hormone receptors (ER and/or PgR) has been reported in five cases [1,2,3,4,5,6,7,8,9,10,11,12,13,14,15,16,17,18,19,20,21,22,23,24,25,26,27,28,29,30,31]. Similarly, in our case, we observed a faint and focal ER positivity. Her2 neu score 2+ has been reported in three cases (two of which with confirmed FISH amplification), and two cases showed Her2 neu score 3+. Thus, a total of four Her2 neu overexpressing cases were described among the known molecular phenotypes [2,3,4,5,6,7,8,9,10,11,12,13,14,15,16,17,18,19,20,21,22,23,24,25,26,27,28,29,30,31]. The Ki-67 proliferation index varied from 3 to 99%, and there were 18 patients with a Ki67 index higher or equal to 30%, including our case [1,2,3,4,5,6,7,8,9,10,11,12,13,14,15,16,17,18,19,20,21,22,23,24,25,26,27,28,29,30,31]. One case had a basal-like immunophenotype, with positivity for CK5/6 and EGFR.

Before rendering a diagnosis of primary MCA of the breast, pathologists should exclude a metastasis from primitive ovarian, pancreatic, and/or gastrointestinal mucinous neoplasms [1,2,3,4,5,6,7,8,9,10,11,12,13,14,15,16,17,18,19,20,21,22,23,24,25,26,27,28,29,30,31,32,33]. Therefore, the clinical history and imaging studies should be integrated with the pathological findings. The morphological identification of an in situ neoplastic component (DCIS) favors a diagnosis of breast primary. Immunohistochemical analyses are an essential tool to achieve a correct diagnosis. It is essential to test TTF1, WT1, PAX8, GATA3, and GCDFP15, which are lineage markers. In this regard, combination of CK7 and CK20 can also be useful: the great majority of breast MCAs are CK7+ and CK20- [33,34,35,36]. However, Chen et al. and Kaur et al. showed the possibility of only focal CK20 positivity in primary breast MCA [5,31]. On the contrary, both CK7 and CK20 are more frequently positive in ovarian and pancreatic mucinous carcinomas. CK20 positivity alone is usually found in gastrointestinal mucinous neoplasms [33]. CDX2 and SATB2 are other useful markers for the differential diagnosis; generally, they are positive in case of gastro-intestinal/pancreato-biliary primitivity and negative in case of mammary primitivity, although, an important pitfall, an occasional weak reactivity for both markers has been reported also in primary breast carcinomas [33,34,35,36]. 

Moreover, various primary breast lesions should be considered in the differential diagnosis of MCA, including mucocele-like lesions, pure mucinous carcinoma, encapsulated papillary carcinoma and invasive papillary carcinoma.

However, the last three usually show strong and diffuse immunoreactivity for ER and PgR [1] and, generally, a low Ki67 proliferation index. Additionally, the presence of abundant intracellular and extracellular mucus helps us to exclude an invasive papillary carcinoma. The presence of neoplastic mucinous cells, the invasive growth, and the loss of myoepithelial component are the main characteristics used to rule out mucocele-like or mucinous cystic lesions.

Consideration should be given regarding the mucin immunohistochemical profile of breast MCA. Mucins are expressed differently according to the organs and tumors types [37,38,39,40,41]. In our case of MCA, the MUC immunohistochemical profile demonstrated a high level of expression of MUC1 and MUC5 and no expression of MUC2 and MUC6, similarly to the case reported by Kim et al. [14]. According to Kim et al. [14], we can confirm a mucin profile, which is quite different from classic mucinous carcinoma arising in the breast and other organs. In the context of a breast mucin-producing carcinoma, a MUC5-positive and MUC2-negative immunoprofile is more suggestive of breast MCA; in contrast, a MUC2-positive and MUC5-negative immunoprofile is more indicative of breast mucinous carcinoma.

Concerning the biological behavior, most reported cases display a relatively favorable prognosis despite a high proliferation index and hormone receptor negativity [33,42], perhaps suggesting that prognosis of this tumor is independent of the molecular subtype.

This tumor rarely invades lymphatic vessels. In fact, axillary lymph node metastases are uncommon, with only six reported patients having lymph node involvement (with no more than three lymph nodes involved). Our case showed ITCs in one axillary lymph node. Distant metastases have only been reported by Kaur K. et al., who described the history of a 65-year-old postmenopausal woman with radiological evidence of lung metastases at the time of presentation [27]. Moreover, Nayak et al. [21] also described a case that recurred eight years after surgery; it was mostly constituted by a DCIS, with only a 5 mm invasive MCA characterized by a triple negative phenotype and the presence of isolated tumor cells in the sentinel lymph node at the time of diagnosis. 

Nevertheless, according to the literature review, only four cases were followed up for more than five years, with only one recently described case with a follow up of 108 months after surgery [31]. The follow-up times of all remaining cases were approximately short in duration (range 1–2 years) [2,3,4,5,6,7,8,9,10,11,12,13,14,15,16,17,18,19,20,21,22,23,24,25,26,27,28,29,30,31]. We conclude that this relatively short follow-up period of observation cannot fully clarify the biological behavior of the tumor. In light of this, MCAs might have a long-term risk of local recurrence. Many more accumulated and well-described cases are needed in order to define a possible metastatic potential and to verify the prognostic factors of MCAs. 

Individualized targeted treatment regimens are still lacking. Surgical resection was performed for all cases, generally in the form of partial or radical mastectomy. Chemotherapy and radiotherapy were chosen in a few cases [2,3,4,5,6,7,8,9,10,11,12,13,14,15,16,17,18,19,20,21,22,23,24,25,26,27,28,29,30,31]. Hormone therapy and HER2 targeted therapy have been adopted, respectively, for hormone receptors positive and HER2+ cases. The unique relapsed case [21] was managed via mastectomy and axillary lymph node dissection alone, without adjunct therapy.

Regarding the molecular landscape, in our research, we found that the genome of only two cases has been described to date. Lin et al. performed a wide molecular profile of a case of primary mucinous cystadenocarcinoma of the breast using next-generation sequencing (NGS) targeting cancer-related genes [28]. They discovered a frameshift deletion in BAP1 and RB genes and missense mutation in TP53 gene (also confirmed via immunohistochemistry), therefore suggesting the driving role of molecular alterations in suppressor genes, mostly those controlling cell cycle, cell proliferation, cell senescence and chromatin remodeling [28]. 

Furthermore, recently, Ting Lei et al. described a case of mammary mucinous cystadenocarcinoma in a 59-year-old woman, adding important molecular information. A total of 425 genes were sequenced using next-generation sequencing technology. The authors reported a gene profile, very close to that of a common high-grade TNBC, in which PIK3CA, TP53, KRAS and RB1 mutations are commonly present. In particular, the authors described a missense mutation in PIK3CA (c.3140A>G, p. H1047R), a common activation mutation in breast cancer, enhancing PI3K lipid kinase activity, stimulating the PI3K/AKT signaling pathway and promoting the tumoral invasion and metastasis. They also reported the hot-spot mutation site of the KRAS gene (c.35G>T, p.G12V), causing impairment of the GP-mediated hydrolytic function of GTP, resulting in increased intracellular RAS-GTP levels and RAS pathway activation. In addition, they underlined the role of MAP2K4 (c.257_258del, p. R86Tfs*7) mutation in mediating cell invasion. Other recurrent mutations in KDR, PKHD1 and TERT genes have been discovered but their significance is still unclear/undefined, and thought to possibly be related to tumoral formation. Finally, Lei et al. referred a relatively high tumor mutational burden (TMB) of 9.27, in absence of a microsatellite instability high (MSI-H) status.

## 4. Conclusions

In this study, we present a primary MCA of the breast with associated invasive lobular carcinoma component. This association has never been reported in the literature.

Primary MCA of the breast is a rare neoplasm, mostly affecting perimenopausal and postmenopausal women. Morphologically, it overlaps with other mucin-producing breast tumours and its counterparts in the pancreas, ovary, or gastrointestinal tract, making the correct diagnosis challenging.

Based on limited studies, the outcome is generally favorable, despite the triple negative immunoprofile and the high reported proliferation index Ki67, considered to be poor prognostic factors in breast cancer.

The clinical history, the presence of adjacent DCIS and/or areas of conventional invasive no special type (NST) carcinoma can help the pathologist to achieve the diagnosis of primary breast MCA. In cases of complete absence of a peri/intralesional DCIS, the diagnosis of a breast MCA could be troubling, requiring the use of a wide panel of immunohistochemical biomarkers to achieve a correct diagnosis. Only the combination of clinical data, imaging studies, morphological and immunohistochemical features is helpful to definitively confirm the pathological diagnosis. Hopefully, an in-depth study of tumoral genomic characteristics could help clinicians predict the prognosis and provide alternative therapeutic tools, such as PIK3CA and KRAS genes, which have well-known corresponding targeted inhibitors.

## Figures and Tables

**Figure 1 jpm-13-00948-f001:**
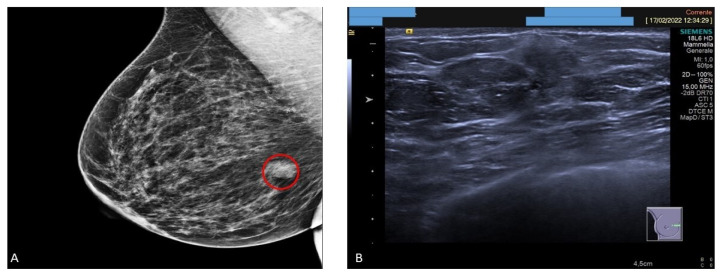
Mammographic and ultrasound features. (**A**) Mammography examination showing a solid nodule in the internal equatorial location (red circle). (**B**) Ultrasound examination demonstrating a solid nodule devoid of vascular structures.

**Figure 2 jpm-13-00948-f002:**
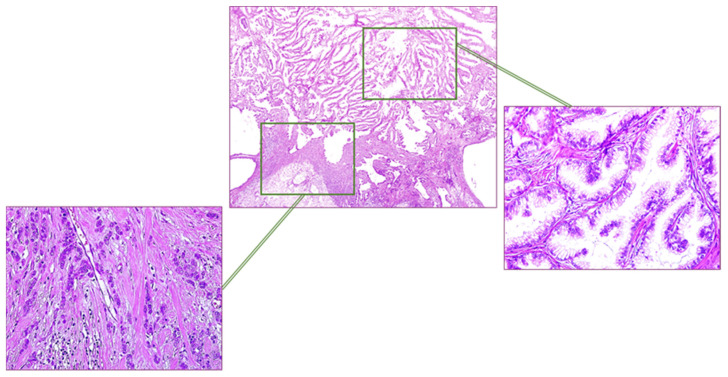
Microscopic features of the surgical sample (hematoxylin and eosin, 2× magnification—**center top**): MCA (**bottom right**, 20× magnification) characterized by tortuous glands lined by columnar cells with basal-located hyperchromatic nuclei, and apical mucin; CLI (**bottom left**, 20× magnification) characterized by chains of medium-sized cells, with large hyperchromatic nuclei.

**Figure 3 jpm-13-00948-f003:**
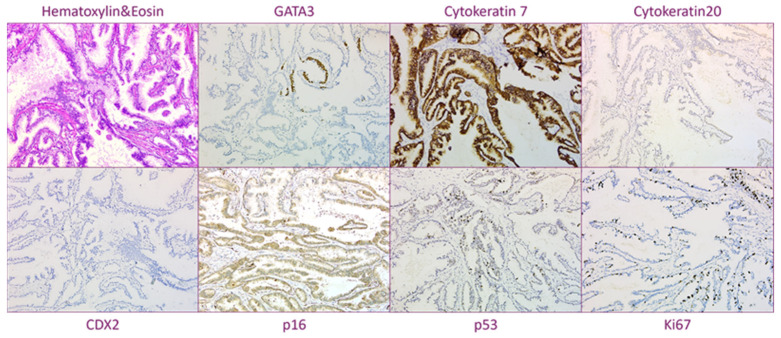
Immunohistochemical profile of MCA (20× magnification). (Normal mammary ducts served as positive internal control for Cytokeratin 7 and Gata 3. Lymphocytes served as positive internal controls for KI67).

**Figure 4 jpm-13-00948-f004:**
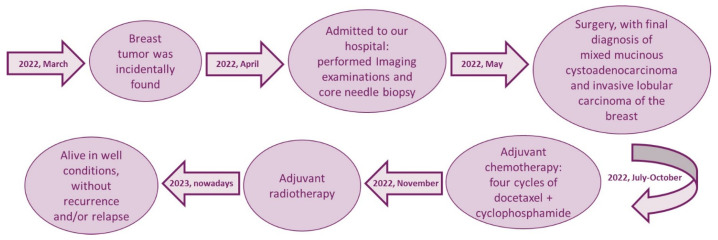
Timeline of clinical history of our patient.

**Table 1 jpm-13-00948-t001:** Literature review. Abbreviations: DCIS Ductal Carcinoma In Situ; DOD Died of Other Disease; ANED Alive with No Evidence of Disease; NA Not Available.

Case	Age (Years)	Size (cm)	DCIS	ER/PR/HER2	CK7/CK20	Nodal Metastasis	Treatment	Follow-Up
1 [1]	79	6	+	NA	NA/NA	-	Mastectomy	DOD, 24 months
2 [2]	54	19	-	-/-/NA	+/-	+	Mastectomy, Lymph node dissection	ANED, 24 months
3 [2]	67	2.3	+	-/-/NA	+/-	-	Mastectomy, Lymph node dissection	ANED, 22 months
4 [2]	49	8.5	+	-/-/NA	+/-	-	Mastectomy, Lymph node dissection, Chemotherapy, Radiotherapy	ANED, 11 months
5 [2]	61	0.8	-	-/-/NA	+/-	-	Lumpectomy, Lymph node dissection	NA
6 [3]	74	10	+	-/NA/NA	+/-	-	Mastectomy, Lymph node dissection	ANED, 24 months
7 [4]	96	2	-	-/-/-	NA/NA	+	Lumpectomy, Lymph node dissection	ANED, 46 months
8 [5]	65	3	+	-/-/-	+/+ (focal)	-	Mastectomy, Lymph node dissection, Chemotherapy	ANED, 8 months
9 [6]	51	4	-	-/-/NA	+/-	-	Lumpectomy	NA
10 [7]	55	2.5	+	-/-/-/	+/-	-	Lumpectomy	ANED, 6 months
11 [8]	52	10	-	+/-/-	-/-	-	Mastectomy, Lymph node dissection	ANED, 24 months
12 [9]	61	3	-	-/-/-	NA/-	-	Mastectomy, Lymph node dissection	ANED, 6 months
13 [10]	73	4.5	+	-/-/2+	+/-	-	Mastectomy, Lymph node dissection	NA
14 [11]	65	3	+	-/-/NA	+/-	+	Partial mastectomy, Lymph node dissection	ANED, 6 months
15 [12]	52	6.5	-	-/-/-	+/-	-	NA	NA
16 [13]	41	7	+	-/-/-	+/-	-	Mastectomy, Lymph node dissection	ANED, 24 months
17 [14]	59	0.9	+	-/-/2+	+/-	-	Partial mastectomy, lymph node dissection, Chemotherapy	ANED, 3 months
18 [15]	62	5.6	-	-/-/-	+/-	-	Mastectomy, Lymph node dissection	ANED, 15 months
19 [16]	55	2	+	-/-/2+	+/-	-	Mastectomy, Lymph node dissection, Chemotherapy, Radiotherapy	ANED, 10 months
20 [17]	91	7.5	+	-/-/-	+/-	-	Mastectomy, Radiotherapy	DOD, 14 months
21 [18]	59	2	-	-/-/3+	NA/NA	-	Partial mastectomy	NA
22 [18]	50	2.2	-	-/-/-	NA/NA	-	Partial mastectomy	NA
23 [19]	63	1.6	-	-/-/-	+/-	+	Lumpectomy, Lymph node dissection, Chemotherapy, Radiotherapy	ANED, 48 months
24 [20]	58	4.5	+	-/-/-	+/-	-	Mastectomy, Lymph node dissection	ANED, 6 months
25 [21]	68	6.2	+	-/-/-	+/-	-	Lumpectomy, Sentinel lymph node dissection	ANED, 3 months
26 [21]	59	2	+	-/-/-	+/-	-	Lumpectomy, lymph node dissection	Alive with local recurrence after 96 months
27 [22]	56	2	-	+/+/-	-/-	-	Mastectomy, Lymph node dissection	ANED, 3 months
28 [23]	68	4	NA	-/-/-	+/-	-	Mastectomy, Lymph node dissection	ANED, 21 months
29 [24]	45	4.3	+	-/-/-	+/-	-	Lumpectomy, Mastectomy, Lymph node dissection, Chemotherapy, Radiotherapy	ANED, 6 months
30 [25]	69	2	+	+/+/-	NA	-	NA	NA
31 [26]	66	2.5	+	-/-/-	+/-	-	Mastectomy, Lymph node dissection	ANED, 13 months
32 [27]	65	18	-	-/-/3+	+/focal +	NA	Mastectomy	ANED, 6 months
33 [28]	72	0.9	-	-/-/-	+/-	-	Mastectomy, Lymph node dissection	ANED, 16 moths
34 [29]	61	3.5	+	+/+/-	+/-	+	Mastectomy, Lymph node dissection	ANED, 8 months
35 [30]	61	2.5	+	-/-/-	+/-	-	Mastectomy, Lymph node dissection	ANED, 10 months
36 [31]	59	3	-	-/-/-	+/-	-	Lumpectomy, Lymph node dissection	ANED, 108 months
Present Case	41	1.5	-	+/-/-	+/NA	Isolated Tumor Cells	Partial mastectomy, sentinel lymph node dissection	ANED, 11 months

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
