# Peer review of "Primary Mucinous Cystadenocarcinoma of the Breast Intermixed with Pleomorphic Invasive Lobular Carcinoma: The First Report of This Rare Association"

_jpm, 2023, doi:10.3390/jpm13060948_

Round 1

Reviewer 1 Report

This manuscript reviews what is known about primary mucinous cystuadenocarcinoma of the breast. In general the report is straight forward and clinically useful.

There are some issues that need to be attended to:

1. The case report needs to be streamlined.

2. In line 176-177, the % ER+ and PR+ cells should be reported.

3. In Figures 2 and 3, the slides are overstrained with Eosin; the slides need to be properly stained. In Figure 3, positive and negative controls should be provided for each antibody; p16 staining is blurry, this should be corrected.

4. In any of these cases, did family members have other mucinous tumors?

5. The discussion should be cut by 50%.

Author Response

Thank you for inviting us to submit a revised draft of our manuscript. We also appreciate the time and effort you and each of the reviewers have dedicated to providing insightful feedback on ways to strengthen our paper. Thus, it is with great pleasure that we resubmit our article for further consideration. We have incorporated changes that reflect the detailed suggestions you have graciously provided.

To facilitate your review of our revisions, the following is a point-by-point response to the questions and comments.

Authors’ point-by-point responses to reviewers comments

Reviewer 1:

This manuscript reviews what is known about primary mucinous cystuadenocarcinoma of the breast. In general the report is straight forward and clinically useful. There are some issues that need to be attended to:

Q1. The case report needs to be streamlined.

A1: We have rewritten great parts of the text, hoping to have made simpler and more effective the text reading

Q2. In line 176-177, the % ER+ and PR+ cells should be reported.

A2: We have added the % values of ER and PR

Q3. In Figures 2 and 3, the slides are overstrained with Eosin; the slides need to be properly stained. In Figure 3, positive and negative controls should be provided for each antibody; p16 staining is blurry, this should be corrected.

A3: We have modified the figures according to reviewer suggestion. Positive internal controls for CK7, GATA 3 and Ki67 have been included in the figure legends.

Q4. In any of these cases, did family members have other mucinous tumors?

A4: We have check the cases reported in the literature review: there are not cases with family history of mucinous tumors

Q5. The discussion should be cut by 50%.

A5: We have done it

Again, thank you for giving us the opportunity to strengthen our manuscript with your valuable comments and queries. We hope that these revisions persuade you to accept our submission.

Finally, we are kindly asking you the possibility to add another Author in the Authors list, the Dr Valeria Masciullo, who has deeply contributed to the revision of the text and to the check and update of the literature review

Sincerely yours,

Dr. Angela Santoro

Reviewer 2 Report

The main question addressed by the research is novelty of specific Mucinous variety of breast malignancy. The reviewer considered the topic original and relevant in the field, and address a specific gap in the field. Compared with other published material, the subject area needs to be differentiated from Mucinous secondary in breast. Methodology is acceptable for case report. The conclusions are consistent with the evidence and arguments presented and they address the main question posed. References are appropriate. The whole article is based on immunohistopathological differentiation of various Mucinous varieties Stress is on keeping in mind the possibility of specific Mucinous primary malignancy of breast and its outcome wrt secondary.

Author Response

Thank you for inviting us to submit a revised draft of our manuscript. We also appreciate the time and effort you and each of the reviewers have dedicated to providing insightful feedback on ways to strengthen our paper. Thus, it is with great pleasure that we resubmit our article for further consideration. We have incorporated changes that reflect the detailed suggestions you have graciously provided.

To facilitate your review of our revisions, the following is a point-by-point response to the questions and comments.

 Authors’ point-by-point responses to reviewers comments

 Reviewer 2

The main question addressed by the research is novelty of specific Mucinous variety of breast malignancy. The reviewer considered the topic original and relevant in the field, and address a specific gap in the field. Compared with other published material, the subject area needs to be differentiated from Mucinous secondary in breast. Methodology is acceptable for case report. The conclusions are consistent with the evidence and arguments presented and they address the main question posed. References are appropriate. The whole article is based on immunohistopathological differentiation of various Mucinous varieties Stress is on keeping in mind the possibility of specific Mucinous primary malignancy of breast and its outcome wrt secondary.

Thank you for your positive and encouraging comments.

Again, thank you for giving us the opportunity to strengthen our manuscript with your valuable comments and queries. We hope that these revisions persuade you to accept our submission.

Finally, we are kindly asking you the possibility to add another Author in the Authors list, the Dr Valeria Masciullo, who has deeply contributed to the revision of the text and to the check and update of the literature review

Sincerely yours,

Dr. Angela Santoro

Round 2

Reviewer 1 Report

please publish in current form